# Optimized Nuclear Pellet Method for Extracting Next-Generation Sequencing Quality Genomic DNA from Fresh Leaf Tissue

**DOI:** 10.3390/mps2020054

**Published:** 2019-06-25

**Authors:** Md Masud Rana, Murat Aycan, Takeshi Takamatsu, Kentaro Kaneko, Toshiaki Mitsui, Kimiko Itoh

**Affiliations:** 1Department of Life and Food Sciences, Graduate School of Science and Technology, Niigata University, Niigata 950-2181, Japan; f16m502g@mail.cc.niigata-u.ac.jp (M.M.R.); takamatsutakeshi@yahoo.co.jp (T.T.); k-neko@gs.niigata-u.ac.jp (K.K.); t.mitsui@agr.niigata-u.ac.jp (T.M.); 2Agronomy Division, Bangladesh Rice Research Institute, Gazipur-1701, Bangladesh; 3Institute of Science and Technology, Niigata University, Niigata 950-2181, Japan; murataycan@agr.niigata-u.ac.jp

**Keywords:** DNA extraction, high-quality gDNA, next-generation sequencing

## Abstract

Next-generation sequencing (NGS) is a revolutionary advancement allowing large-scale discovery of functional molecular markers that has many applications, including plant breeding. High-quality genomic DNA (gDNA) is a prerequisite for successful NGS library preparation and sequencing; however, few reliable protocols to obtain such plant gDNA exist. A previously reported nuclear pellet (NP) method enables extraction of high-yielding gDNA from fresh leaf tissue of maize (*Zea mays* L.), but the quality does not meet the stringent requirements of NGS. In this study, we optimized the NP method for whole-genome sequencing of rice (*Oryza sativa* L.) through the integration of simple purification steps. The optimized NP method relied on initial nucleus enrichment, cell lysis, extraction, and subsequent gDNA purification buffers. The purification steps used proteinase K, RNase A, phenol/chloroform/isoamyl alcohol (25:24:1), and chloroform/isoamyl alcohol (24:1) treatments for protein digestion and RNA, protein, and phenol removal, respectively. Our data suggest that this optimized NP method allowed extraction of consistently high-yielding and high-quality undegraded gDNA without contamination by protein and RNA. Moreover, the extracted gDNA fulfilled the quality metrics of NGS library preparation for the Illumina HiSeq X Ten platform by the TruSeq DNA PCR-Free Library Prep Kit (Illumina). We provide a reliable step-by-step guide to the extraction of high-quality gDNA from fresh leaf tissues of rice for molecular biologists with limited resources.

## 1. Introduction

Next-generation sequencing (NGS) is an advancement in molecular biology that allows the detection of thousands of nucleotide variations and the development of functional molecular markers in various plant species regardless of their genome size [1]. These markers have been used successfully in plant breeding to improve economically important agronomic traits from multiple crop species. NGS involves several high-throughput sequencing technologies, such as Illumina, Ion Torrent, PacBio, and Oxford Nanopore [1]. Library preparation, sequencing runs, sequence validity, and the number of reads in these technologies depend mostly on the purity and quality of the genomic DNA (gDNA) [2,3]. In recent years, NGS technologies have become popular with molecular biologists due to their low cost, high accuracy, and reliability. The gDNA for NGS must meet strict quality control (QC) requirements, including high molecular weight; an intact and undegraded structure upon gel electrophoresis; an *A*_260_/*A*_280_ ratio of 1.8–2.0 (indicating purity); and a lack of contamination from secondary metabolites, phenols, and RNA [2,4,5]. Using undegraded DNA is the key to obtaining longer sequencing reads in third-generation sequencers [6,7]. The isolation of high-purity gDNA also improves the accuracy and cost efficiency of high-throughput sequencing. 

The hexadecyltrimethylammonium bromide (CTAB) method is a widely used gDNA extraction protocol [8] that is applicable to diverse samples. In the last few years, a significant number of gDNA extraction methods from plant tissues have been published [2,3,7,9,10,11,12,13]. Some of these either modified or improved the traditional CTAB-based method, targeting improved yields and quality. There are, however, few published reports demonstrating the efficiency of gDNA extraction procedures from plant samples for NGS applications. In addition, the isolation of gDNA for downstream genetic analysis using the available commercial kits is expensive and not suitable for all species. However, reliable protocols to obtain high-quality plant genomic DNA are few, impeding NGS technologies and further functional genomics research. The quality of gDNA and its integrity are crucial for most downstream applications in PCR [14] and high-throughput technologies [2,3]. The purity of DNA varies according to the extraction method and tissue source [15]. gDNA purification from plants can be problematic due to contamination from proteins, polysaccharides, residual phenols, and ribonucleotides. The gDNA could be purified from these contaminants through digestion and precipitation using proteinase K, a mixture of phenol and chloroform, and RNase A treatments. Many commercial gDNA purification kits are also available and suitable for a variety of applications.

The nuclear pellet (NP) method enables the extraction of high-yielding gDNA from fresh leaf tissues of maize (*Zea mays* L.) and relies mostly on grinding, cell lysis, and DNA precipitation buffers [16]. The gDNA of the NP method is suitable for PCR-based assays [17] but has not been sufficiently optimized for NGS analysis. 

Here, we optimized the NP method for extraction of rice (*Oryza sativa* L.) gDNA by integrating some simple purification steps. We assessed and compared the yield, quality, and integrity of extracted DNA using the optimized NP method with the CTAB-based QIAGEN DNeasy Plant Mini Kit and NP protocols, using fluorometric and qubit measurements and gel electrophoresis. Furthermore, the gDNAs from the optimized NP method were sequenced on the Illumina Hiseq X Ten platform with NGS libraries prepared by the TruSeq DNA PCR-Free Library Prep Kit (Illumina) to evaluate the quality of the sequenced data.

## 2. Materials and Methods

### 2.1. Plant Materials

Seeds of the rice variety “Yukinko-mai” were sterilized with 2.5% sodium hypochlorite, rinsed with water, and incubated at 28 °C for two days in the dark. Pregerminated seeds were then grown in the seedling tray. Twenty-day-old rice seedlings were used for the extraction of gDNA.

### 2.2. Genomic DNA Extraction

#### 2.2.1. CTAB-Based Method

Genomic DNA was extracted using the CTAB-based method [8] with some modifications. Fresh leaf samples (100 mg) were frozen in liquid nitrogen and ground to a fine powder using a mortar and pestle. The CTAB extraction buffer (1.4 M NaCl, 100 mM Tris-HCl pH 8.0, 20 mM EDTA pH 8.0, 2% CTAB) was then added. After incubation at 65 °C for 1 h, purification with phenol:chloroform:isoamyl alcohol (25:24:1) and DNA precipitation with isopropanol was performed. The DNA pellet was washed with 1 mL of 70% ethanol and air-dried. Finally, the DNA pellet was resuspended in 100 µL Milli Q water. After suspension, 1 µL of RNase A (10 ng) was added and incubated for 30 min at 37 °C. The experiment used three biological replicates, each with a 100 mg leaf sample, and the final gDNA concentration was calculated for a 1 g sample.

#### 2.2.2. Qiagen Method

Genomic DNA from rice leaf tissue (100 mg) was extracted using the DNeasy Plant Mini Kit (Qiagen, Valencia, CA, USA) according to the manufacturer’s protocol. The experiment used three biological replicates, each with a 100 mg leaf sample, and the final gDNA concentration was calculated for a 1 g sample.

#### 2.2.3. Optimization of Nuclear Pellet Method

##### Reagents


Solution I: 15% sucrose, 50 mM Tris-HCl (pH 8.0), 50 mM EDTA (pH 8.0), and 500 mM NaClResuspension buffer: 20 mM Tris-HCl (pH 8.0) and 10 mM EDTA (pH 8.0)20% sodium dodecyl sulfate (SDS)7.5 M ammonium acetate5 M NaClIsopropanolTE buffer: 10 mM Tris-HCl (pH 7.6) and 0.1 mM EDTARNase A: a working solution of 1 µg/µL (final concentration ≈ 10 ng/µL)Proteinase K: a working solution of 2 µg/µL (final concentration ≈ 80 ng/µL)PCI: phenol/chloroform/isoamyl alcohol (25:24:1)CIA: chloroform/isoamyl alcohol (24:1)100% ethanol (*v*/*v*)70% ethanol (*v*/*v*)Qubit^TM^ dsDNA HS Assay Kit (Thermo Fisher Scientific Inc., Waltham, MA, USA)


##### Equipment


Mortar and pestleDry block heaterTable-top high-speed microcentrifugeFreezer (−20 °C)Gel electrophoresis systemEppendorf BioSpectrometer^®^ fluorescence (Eppendorf AG, Hamburg, Germany)


Note: Prepare all reagents using Milli Q water; autoclave glassware before use.

##### Protocol

The protocol for gDNA extraction using the optimized NP method is as follows and is shown schematically in Figure 1:Weigh 1 g of young fresh leaf tissue and cut into small pieces using clean and sharp scissors. Place the cut tissue in liquid nitrogen and grind thoroughly with a mortar and pestle. Add 12 mL of Solution I to the fine tissue powder and suspend well. Transfer the suspended tissue (2 mL in each) into six individual 2 mL microcentrifuge tubes.Centrifuge the tubes containing the suspended tissue at 500 rpm for 3 min at 4 °C. Discard the upper phase carefully. Centrifuge at 1000 rpm for 1 min at 4 °C and discard the upper phase.Add 450 µL of resuspension buffer and 30 µL of 20% SDS into each tube containing tissue, shake briefly, and incubate at 70 °C for 15 min.Add 230 µL of 7.5 M NH_4_OAc into each tube, shake vigorously, and incubate the reaction mixture on ice for at least 30 min.Centrifuge the reaction mixture at 15,000 rpm for 20 min at 4 °C and divide equal amounts of the cleared supernatant into six individual microcentrifuge tubes, labeled #1–6. Repeat this step (once).Add 1 volume of isopropanol to the supernatant of each tube, gently mix by inverting, and centrifuge at 15,000 rpm for 15 min at 25 °C. Discard the supernatant and air-dry pellet for 5–10 min (do not excess dry).Add 50 µL of TE buffer individually to tubes #1–5 and wait until the pellet has dissolved. Transfer the dissolved DNA from tubes #1–5 into tube #6. Add 50 µL TE buffer to the original tube #1, allow to dissolve, then transfer it to tube #2. Repeat this transfer and dissolving process until tube #5, then transfer the DNA to tube #6. The final volume of the DNA sample in tube #6 will be 300 µL.Add 600 µL of 100% ethanol to the sample in tube #6, centrifuge at 15,000 rpm for 10 min at 25 °C, and decant the supernatant. Again, add 600 µL of 100% ethanol to the DNA pellet in tube #6, centrifuge at 15,000 rpm for 1 min, and decant the supernatant.Air-dry the pellet for 5–10 min. Resuspend the pellet in 50 µL of TE buffer (do not vortex).Add 0.5 µL of RNase A to the solution and incubate at 37 °C for 1 h.Add 2.2 µL of proteinase K to the solution and incubate at 37 °C for 1 h.Check gDNA quality by electrophoresis using a 0.7% agarose gel (optional).Add 400 µL of TE buffer to the digested sample containing the DNA and mix gently by pipetting.Add 450 µL of PCI to the DNA solution, gently mix, and centrifuge at 15,000 rpm for 15 min at 4 °C. Transfer the cleared supernatant to a new 1.5 mL collection tube. Repeat this step (once).Add 450 µL of CIA to the DNA solution, mix gently, and centrifuge at 15,000 rpm for 15 min at 4 °C. Transfer the cleared supernatant to a new 1.5 mL collection tube. Repeat this step (once).Add 27 µL of 5 M NaCl and 1 mL of 100% ethanol to the solution, gently mix, and incubate at −20 °C for 1 h.Centrifuge at 15,000 rpm for 15 min at 4 °C and discard the supernatant.Wash the pellet with 100 µL of 70% ethanol, centrifuge at 15,000 rpm for 5 min, and discard the ethanol.Wash the pellet with 100 µL of 100% ethanol, centrifuge at 15,000 rpm for 5 min, and discard the supernatant. Again, add 100 µL of 100% ethanol to the pellet and wash and discard the ethanol.Air-dry the pellet for 5–10 min. Resuspend the pellet in 11 µL of TE buffer.Dilute the DNA with TE buffer (as required) for downstream analysis.

Note: Steps 1–9 correspond to the NP method of crude genomic DNA extraction.

### 2.3. DNA Quantification

The quantity of genomic DNA isolated by each method was tested with a Qubit^TM^ dsDNA HS Assay Kit, following the manufacturers’ procedures.

### 2.4. DNA Quality Assessment

The quality of extracted DNA was determined at 260 nm using the Eppendorf BioSpectrometer^®^ fluorescence with 1 µL of each sample, as described by the manufacturer. Agarose gel electrophoresis was used to further assess the gDNA quality in 0.8% agarose gel, visualizing the DNA using 0.5 µg/mL ethidium bromide in Tris/Borate/EDTA (TBE).

### 2.5. Quantification of Nuclear, Chloroplast, and Mitochondrial DNAs by qPCR

The nuclear, chloroplast, and mitochondrial DNAs were quantified by quantitative real-time PCR (qPCR) according to our previous protocol [18]. The qPCR was performed using SsoFast EvaGreen Supermix (Bio-Rad) on a CFX96 real-time PCR system/C1000 Thermal Cycler (Bio-Rad). Two nanograms of plant DNA from each extraction method were used in a 10 μL reaction mixture. The thermocycling conditions were denaturation at 98 °C for 2 min and 39 cycles of 98 °C for 2 s and 60 °C for 5 s. All the primer sets used to analyze the genome copy number are listed in Appendix A. First, we calculated the copy ratios of nuclear DNA/chloroplast DNA genome and nuclear DNA/mitochondrial DNA genome by relative quantification (2^−ΔΔct^ method), and then each genome DNA content was quantified from the copy ratio and genome size (nuclear, 373,245,519 bp; plastid, 134,525 bp; mitochondrial, 490,520 bp).

## 3. Results and Discussion

In the current study, we optimized the existing nuclear pellet method [16] to improve gDNA purity for NGS library preparation (Figure 1). The optimization was based mainly on the incorporation of simple purification steps, such as proteinase K, RNase A, PCI, and CIA treatments for protein digestion and RNA, protein, and phenol removal, respectively. We added RNase A and proteinase K before the PCI and CIA extractions to obtain highly purified gDNA. 

Our protocol resulted in a mean yield of 10.40 µg gDNA/g fresh tissue (Table 1), significantly higher than the CTAB-based and QIAGEN DNeasy Plant Mini Kit extraction protocols. Due to the high gDNA yield, this protocol allows a downscaling of the amounts of leaf tissue and reagents used. The extracted gDNA obtained from the optimized protocol had an *A*_260_/*A*_280_ ratio of 1.87 and the absorbance showed a single peak at 260 nm (Table 1, Figure 2A), indicating that the gDNA was pure and contamination free. 

The integrity of the gDNA isolated by each method was also evaluated using 0.8% agarose gel electrophoresis. The gDNA obtained from the optimized NP method gave a clear, high-molecular-weight band with little or no smearing (Figure 2B). In contrast, the gDNAs of the CTAB-based, QIAGEN DNeasy Plant Mini Kit, and NP methods showed substantial smearing, indicating degradation of gDNA. The above results clearly demonstrate that improved DNA yield, quality, and integrity were obtained through optimization of the NP method. To purify the genomic DNA, the optimized NP method required 2 h more than the NP method (Appendix A). 

We used qPCR to determine the genome relative abundance in each extraction. Although the optimized NP method slightly increased the nuclear DNA purity, the extracted DNA still contained the high copy number of organelle DNA (chloroplast and mitochondrial DNA), suitable for general NGS analysis (Figure 3A,B).

Furthermore, we extracted DNAs from Yukinko-mai and introgressed rice plants using the optimized NP method for whole-genome sequencing [19]. The genomic DNA samples were sent to Macrogen Japan (http://www.macrogen-japan.co.jp/), where both the samples passed the stringent QC requirements (Table 2). The gDNAs were successfully sequenced on the Illumina Hiseq X Ten platform with NGS libraries prepared by the TruSeq DNA PCR-Free Library Prep Kit (Illumina, Inc., San Diego, CA, USA). In total, we obtained 200,938,541 (Yukinko-mai, DRR151852) and 232,059,326 (introgressed line, DRR151851) paired-end 150 bp reads (Figure 4). 

We removed low-quality reads and trimmed low-quality bases using Trimmomatic v. 0.33 software [20] and assessed the quality of our sequenced data. Figure 4 shows that Phred quality scores (which indicate base calling accuracy) were high even in the latter half of read2, resulting in a high number of pairs of reads surviving the QC process. This result indicates that the optimized NP method provides accurate base calling and high read yields in the Illumina sequencing platform. These days, long-read sequencers such as PacBio RSII/Sequel and Oxford Nanopore Technologies provide powerful applications in de novo sequencing assembly and accurate structural variant (SV) analysis [21,22,23]. They also resolve breakpoints and repetitive regions that are insufficiently characterized using short-read sequencing technologies [24]. In addition, Linked Read Sequencing (10X Genomics Chromium Technology) is capable of obtaining haplotype-phased genome assembly, SNP/INDELs, SVs, and copy number variants (CNVs) [22,25]. These technologies require high-quality intact DNA of high molecular weight [6,7]. We consider that the gDNA obtained using our modified NP protocol fulfills the stringent requirements of these sequencing platforms better than conventional DNA extraction methods and will improve the analytical genomics of any plant species.

## 4. Conclusions

Our optimized nuclear pellet method is very efficient and reliable in extracting high-quality gDNA and is appropriate for targeted NGS library preparation followed by sequencing. This report provides a reliable step-by-step guide for molecular biologists on the isolation of high yields of pure gDNA from fresh leaf tissues of rice.

## Figures and Tables

**Figure 1 mps-02-00054-f001:**
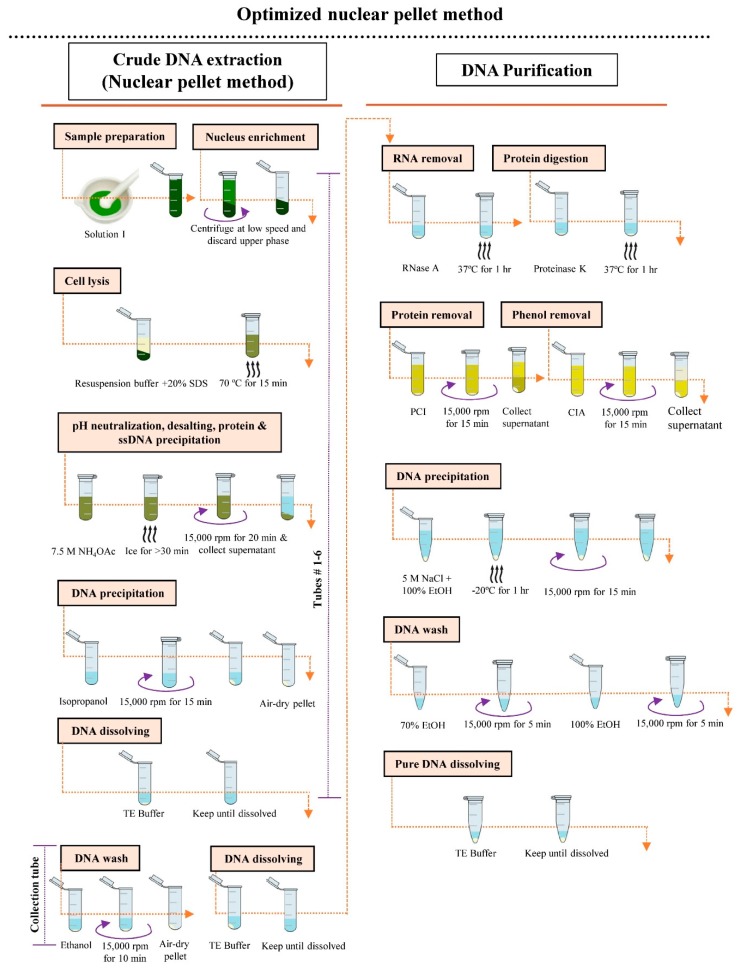
Schematic illustration of the procedures of genomic DNA extraction using the optimized nuclear pellet (NP) method.

**Figure 2 mps-02-00054-f002:**
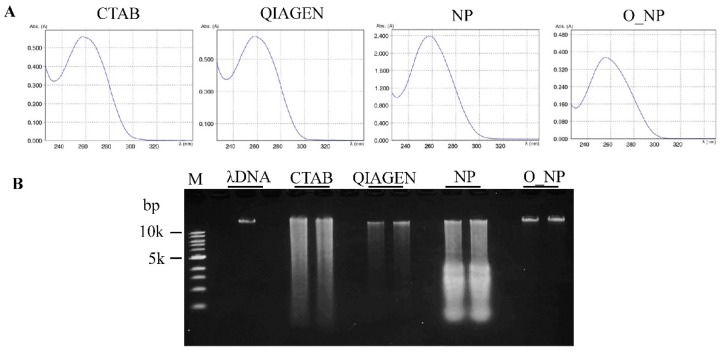
Genomic DNA quality assessment. Genomic DNA from young leaves of rice (*Oryza sativa* L.) plant was extracted by the traditional hexadecyltrimethylammonium bromide (CTAB)-based (CTAB), QIAGEN DNeasy Plant Mini Kit (QIAGEN), NP, and optimized NP (O_NP) methods. (**A**) Comparison of the absorbance profile of genomic DNA at different wavelengths by Eppendorf BioSpectrometer^®^ fluorescence. (**B**) Comparison of genomic DNA integrity using agarose gel electrophoresis. DNA was resolved by electrophoresis in a 0.8% agarose gel and visualized using an ethidium bromide stain. M represents 1 kb DNA ladder (Toyobo Co., Ltd, Osaka, Japan).

**Figure 3 mps-02-00054-f003:**
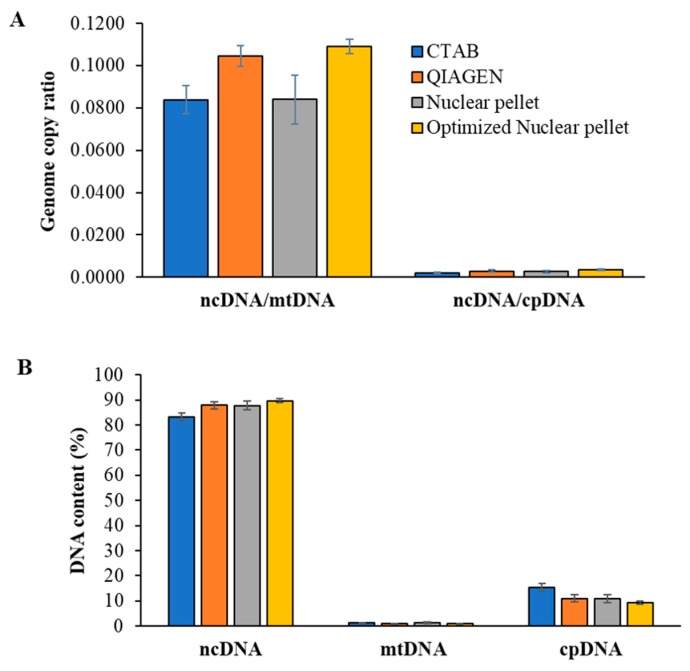
Comparison of genome relative abundance using qPCR. Genomic DNA from young leaves of rice (*Oryza sativa* L.) plant was extracted by the traditional CTAB-based (CTAB), QIAGEN DNeasy Plant Mini Kit (QIAGEN), NP, and optimized NP methods. Genome copy ratio (**A**) and DNA contents (**B**) were assessed according to Takamatsu et al. [18]. ncDNA, mtDNA, and cpDNA represent nuclear DNA, mitochondrial DNA, and chloroplast DNA, respectively. Data are presented as mean ± SD of three independent biological replicates from each extraction method.

**Figure 4 mps-02-00054-f004:**
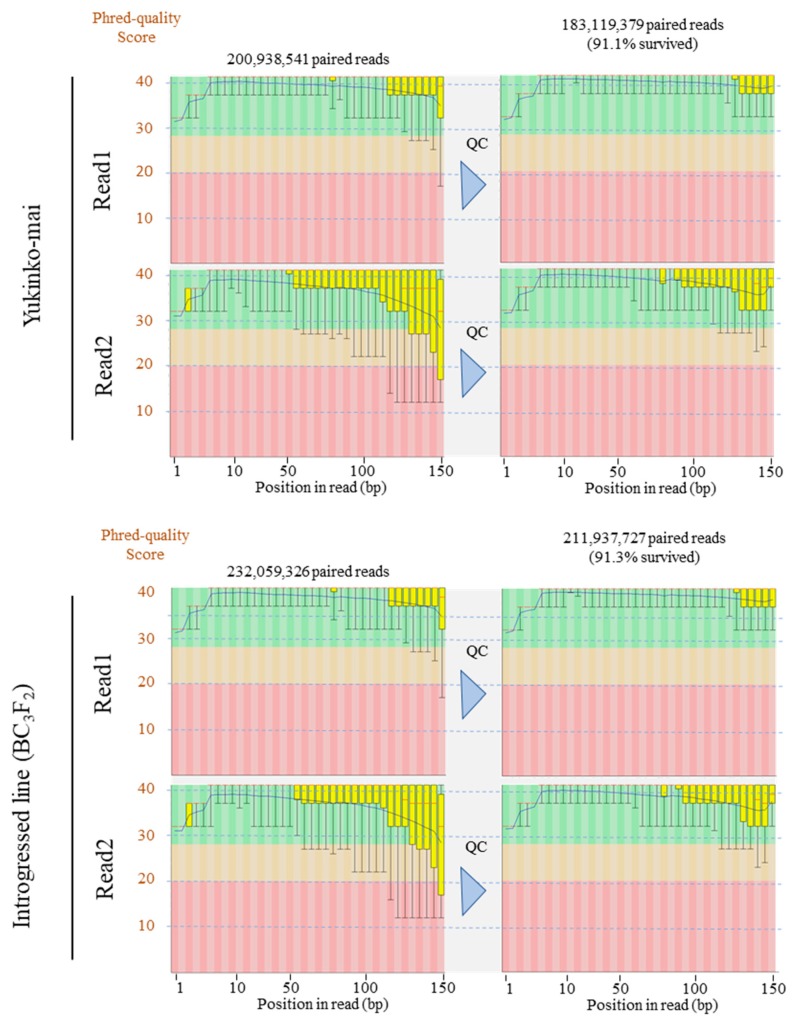
Per base sequence quality of Illumina sequencing reads. Next-generation sequencing (NGS) library was constructed using the gDNA from young leaves of rice (*Oryza sativa* L.) plant extracted via optimized NP protocol. Phred quality scores across paired-end reads before (**left columns**) and after (**right columns**) QC are shown in box plots. The blue and red lines represent the mean quality and median value, respectively. The yellow box represents the first quartile and third quartile. The whiskers extend from the ends of the box to the top 10% and 90% values. The background of the graph in the *y*-axis shows sequence quality. Green, very good quality calls; orange, reasonable quality; red, poor quality. Quality control was performed using Trimmomatic v. 0.33 software [20] with the following parameters: SLIDINGWINDOW: 8:20; TRAILING: 30; MINLEN: 70.

**Table 1 mps-02-00054-t001:** Comparison of the genomic DNA absorbance and yield of rice (*Oryza sativa* L.) tissue in the four different isolation protocols.

Experimental Method	*A*_260_/*A*_280_ Ratio	*A*_260_/*A*_230_ Ratio	DNA Yield (µg/g Fresh Tissue)
CTAB-based	2.06 ± 0.10	2.39 ± 0.04	6.59 ± 0.80 c
QIAGEN DNeasy Plant Mini Kit	1.91 ± 0.03	2.31 ± 0.03	9.03 ± 1.50 b
Nuclear pellet	2.08 ± 0.01	2.46 ± 0.02	11.78 ± 1.38 a
Optimized nuclear pellet	1.87 ± 0.02	2.26 ± 0.05	10.40 ± 1.32 ab

DNA absorbance was determined by Eppendorf BioSpectrometer^®^ fluorescence. Quantification of DNA was assessed using the Qubit^TM^ dsDNA HS Assay Kit. Data are presented as mean ± SD of three biological replicates. Values with the same letter within a column are not statistically different (Duncan’s multiple range test, *p* < 0.05).

**Table 2 mps-02-00054-t002:** Summary of DNA quality control (QC) results from Macrogen.

Sample Name	DNA Submitted (ng/µL TE Buffer)	Total DNA Submitted (µg)	NGS Quality Control Results
Yukinko-mai	330.7	5.29	Passed
Introgressed line (BC_3_F_2_)	256.8	4.11	Passed

DNA was extracted from young leaves of rice (*Oryza sativa* L.) plant. DNA QC based on measurement of absorbance (*A*_260_/*A*_280_ ratio), quantification of DNA using Picogreen method using Victor X2^®^ fluorometry, and 1.0% agarose gel electrophoresis. Passed: Samples proceeded to the library construction step automatically.

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
