# Peer review of "Optimized Nuclear Pellet Method for Extracting Next-Generation Sequencing Quality Genomic DNA from Fresh Leaf Tissue"

_mps, 2019, doi:10.3390/mps2020054_

Round 1

Reviewer 1 Report

In the manuscript, entitled “Optimized nuclear pellet method for extracting next-generation sequencing quality genomic DNA from fresh leaf tissue”, Rana et al., reported an optimized protocol for extracting DNA from plants. I think the protocol does improve DNA extraction quality from plant cells, and can be very useful. Therefore, I support publication. I have two comments below. However, I am a computational biologist and cannot validate the protocol myself. Consultation with another expert in the field may be useful.

First, in my opinion, figure 2b is very convincing that the protocol significantly improves DNA extraction quality.

Second, although there is significant improvement in DNA extraction quality, I think even conventional methods that authors present in the manuscript should work for typical short-read sequencing, e.g., Illumina Hiseq sequencing, without any problem. However, the authors’ protocol may be very useful when requiring longer sequencing reads, such as long-read sequencing (e.g., nanopore sequencing), and de novo sequencing (e.g., mate-pair sequencing). The authors may want to discuss a little more about the applications of the protocol.

Author Response

Response to Reviewer#1

We would like to thank the reviewer for careful and thorough reading of this manuscript and for the thoughtful comments and constructive suggestions, which help to improve the quality of this manuscript. Our response follows (the reviewer’s comments are in italics).

Second, although there is significant improvement in DNA extraction quality, I think even conventional methods that authors present in the manuscript should work for typical short-read sequencing, e.g., Illumina Hiseq sequencing, without any problem. However, the authors’ protocol may be very useful when requiring longer sequencing reads, such as long-read sequencing (e.g., nanopore sequencing), and de novo sequencing (e.g., mate-pair sequencing). The authors may want to discuss a little more about the applications of the protocol.

Answer: Following your suggestion, we have added following sentences.

Line 209-212 (original version)

“Furthermore, the gDNA obtained using our modified NP protocol fulfills the stringent requirements of the third-generation sequencers more closely than conventional DNA extraction methods, and it provides a powerful analytical opportunity for long-read sequencers.”

Line 264-274 (in the revised version)

“These days, long-read sequencers such as PacBio RSII/Sequel and Oxford Nanopore Technologies provide powerful applications in de novo sequencing and accurately structural variant (SV) analysis [20-22]. They also resolve breakpoints and repetitive regions where are insufficiently characterized using short-read sequencing technologies [23]. In addition, Linked Read Sequencing (10X Genomics Chromium Technology) is capable of obtaining haplotype-phased genome assembly, SNP/INDELs, SVs and Copy Number Variants (CNVs) [21,24]. These new sequencing technologies is sensitive to DNA quality, especially high-molecular weight DNA without smear is key to above applications [6,7]. We consider that the gDNA obtained using our modified NP protocol fulfills the stringent requirements of these sequencers more closely than conventional DNA extraction methods, and it provides a powerful analytical opportunity for rice genomics.”

Reviewer 2 Report

The MS is interesting, I have some concerns that authors can address while revising the MS.

Abstract

In the abstract better to provide the snapshot/principle of the method used in the present study and how and why it produced better results need to be stated in the abstract. 

Introduction

Introduction mostly briefed about the DNA extraction but provided very less information about subsequent DNA purification methods.

Materials methods

Need to make very clear whether the NP-based method followed after the CTAB based DNA extraction

Discussion

Throughout the MS authors claimed for the suitability of the method for NGS based marker genotyping however such application does not need the quality of DNA expected with such a lengthy method. 

Authors also need to discuss the time comparison for different methods used. I also recommend for the in-depth analysis with the Bioanalyser and better to propose the method for whole genome sequencing through longer reads like PacBio 

Please take care of typoes and the grammatical errors at several instances. 

Author Response

Response to Reviewer#2

We would like to thank the reviewer for careful and thorough reading of this manuscript and for the thoughtful comments and constructive suggestions, which help to improve the quality of this manuscript. Our response follows (the reviewer’s comments are in italics).

Q1 “In the abstract better to provide the snapshot/principle of the method used in the present study and how and why it produced better results need to be stated in the abstract”

Answer: Following your suggestion, we added the following statements in the Abstract.

Line 24-27 (in the revised version)

The optimized NP method relies on initial nucleus enrichment, cell lysis, extraction, and subsequent gDNA purification buffers. The purification steps used Proteinase K, RNase A, PCI, and CIA treatments for protein digestion, and RNA, protein and phenol removal, respectively.

Q2 “Introduction mostly briefed about the DNA extraction but provided very less information about subsequent DNA purification methods”.

Answer: Following your suggestion, we added the following statement in the introduction section

Line 62-66 (in the revised version)

 The gDNA purification from plants can be problematic due to contamination from proteins, residual phenols and ribonucleotides (Anderson et al. 2018). The gDNA could be purified from these contaminants through digestion and precipitation using Proteinase K, a mixture of phenol and chloroform and, RNase A treatments. A variety of commercial gDNA purification kits are also available and suitable for a variety of applications.

Q3 “Need to make very clear whether the NP-based method followed after the CTAB based DNA extraction”

Answer: We did not optimized CTAB and QIAGEN methods for gDNA extraction [please see section 2.2.1 and 2.2.2]. We showed it in line 84-89. It clearly shows DNA purification of CTAB method consist of PCI, DNA precipitation with isopropanol, 70% EtOH wash. The experimental design section [section 2.3, Figure 1] shows the optimized NP method.

Q4 “Throughout the MS authors claimed for the suitability of the method for NGS based marker genotyping however such application does not need the quality of DNA expected with such a lengthy method”. 

Answer: The extraction protocol influenced the quality of gDNA and, in turn, the sequencing quality. NGS Library prepared using low quality gDNA (smear and contaminated) results inconsistent and unreliable SNP data (please see reference 3).

Q5 “Authors also need to discuss the time comparison for different methods used.

Answer: Following your suggestion, we have summarized the time comparison for different methods used in the new table (Supplementary Table 1).

Supplementary Table 1: Times required for CTAB-based, QIAGEN DNeasy Plant Mini Kit and Optimized nuclear pellet methods of extracting genomic DNA. The time required was calculated with 4 samples for each extraction method.

Methods

Extraction

Purification

Total (min)

Incubation

 Centrifuge

Incubation

 Centrifuge

CTAB-based

80

35

65

10

190

QIAGEN DNeasy Plant Mini Kit

15

7

5

3

30

Optimized nuclear pellet

185

50

70

55

360

Q6 I also recommend for the in-depth analysis with the Bioanalyser and better to propose the method for whole genome sequencing through longer reads like PacBio” 

Answer: As gDNA is high-molecular weight, we cannot separate with the general Bioanalyzer 2100 – maximam size 12 kbp (Agilent DNA12000 kit) – available in our Lab. Nonetheless, to our knowledge, the Agilent 4200 TapeStation system – sizing range 200 to > 60,000 bp – provides accurate quantification and sizing data than the agarose gel electrophoresis. Unfortunately, we don’t have access yet to this system. Following your suggestion, we have added following sentences for discussing the applications of this protocol.

Line 209-212 (Original version)

“Furthermore, the gDNA obtained using our modified NP protocol fulfills the stringent requirements of the third generation sequencers more closely than conventional DNA extraction methods, and it provides a powerful analytical opportunity for long-read sequencers.”

Line 264-274 (in the revised version)

“These days, long-read sequencers such as PacBio RSII/Sequel and Oxford Nanopore Technologies provide powerful applications in de novo sequencing and accurately structural variant (SV) analysis [20-22]. They also resolve breakpoints and repetitive regions where are insufficiently characterized using short-read sequencing technologies [23]. In addition, Linked Read Sequencing (10X Genomics Chromium Technology) is capable of obtaining haplotype-phased genome assembly, SNP/INDELs, SVs and Copy Number Variants (CNVs) [21,24]. These new sequencing technologies is sensitive to DNA quality, especially high-molecular weight DNA without smear is key to above applications [6,7]. We consider that the gDNA obtained using our modified NP protocol fulfills the stringent requirements of these sequencers more closely than conventional DNA extraction methods, and it provides a powerful analytical opportunity for rice genomics.”

Q7 “Please take care of typoes and the grammatical errors at several instances”. 

Answer:  Done as requested

Reviewer 3 Report

Based on the current state of the manuscript, I am not convinced that the presented optimized plant gDNA extraction method is any better than the already existing DNA extraction methods for NGS.

Several specific comments:

1. Introduction: Lines 44-52 - the whole paragraph is confusing and contains contradictory statements.

2.    All three methods described in the paper (CTAB, QIAGEN and the optimized protocol) are nuclear pellet methods, correct?  No reason to distinguish the optimized third protocol as a nuclear pellet method.

3.    In the experimental design, present and compare all 3 methods (CTAB, QIAGEN and the optimized protocol) step by step and highlight/specify the proposed optimizing modifications.

4.    SDS – write out (Line 89).

5.    Qubit is the trademark name and is used to specify QubitTM fluorometer measurement, there are no “fluorometric and qubit measurements”. 

6.    Figure 1 – needs to be moved to the Experimental design section with highlighted section that were optimized.

7.    Results section: remove lines 164-167 (move into Introduction or Discussion)

8.  Table 1 – no significant difference in yield increase compared to QIAGEN method. 

9.  Page 6, line 190 – I don’t see a symmetrical peak in Fig 2A for 'O_NP'  part of it. It is actually more asymmetrical compared to 'CTAB' or 'QIAGEN'

10. Explain 'After cleaning up using Trimmomatic'  - what did you clean and how?

11.  Table 2 – need to add comparison with CTAB and QIAGEN methods.

12.  Figure 3 - need to perform NGS using gDNA obtained via CTAB and QIAGEN methods and compare the sequence quality results for all 3 DNA extraction methods. Currently there is no comparison and no solid evidence that this optimized protocol is any better than QIAGEN’s or any other.

13.  Page 8, line 238 – explain parameters: SLIDINGWINDOW, TRAILING, MINLEN.

16.  Show the evidence that this optimized protocol allows to save on cost of reagents and amounts of leaf tissue used (Lines 181-182).

Author Response

Response to Reviewer#3

We would like to thank the reviewer for careful and thorough reading of this manuscript and for the thoughtful comments and constructive suggestions, which help to improve the quality of this manuscript. Our response follows (the reviewer’s comments are in italics).

Q1. Introduction: Lines 44-52 - the whole paragraph is confusing and contains contradictory statements.

Answer: We adopted this paragraph (lines 52-60, revised version) from the following research article:

Healey et al., 2014; Abdel‑Latif et al., 2017 [ 2, 5]

Q2.    All three methods described in the paper (CTAB, QIAGEN and the optimized protocol) are nuclear pellet methods, correct?  No reason to distinguish the optimized third protocol as a nuclear pellet method.

Answer: Here we reported three independent gDNA extraction protocols (CTAB, QIAGEN and Optimized nuclear pellet method).

Q3.    In the experimental design, present and compare all 3 methods (CTAB, QIAGEN and the optimized protocol) step by step and highlight/specify the proposed optimizing modifications.

We did not optimized CTAB and QIAGEN methods for gDNA extraction [please see section 2.2.1 and 2.2.2]. The experimental design section [section 2.3, Figure 1] clearly shows the optimization of NP method.

Q4.    SDS – write out (Line 89, original version).

Answer:  Done as suggested

         20 % SDS       20 % Sodium dodecyl sulfate

Q5.    Qubit is the trademark name and is used to specify QubitTM fluorometer measurement, there are no “fluorometric and qubit measurements”. 

Answer: Following your suggestions have corrected the DNA quantification and DNA quality assessment sections

Quantify the DNA isolated by each method using QubitTM dsDNA HS Assay Kit, following the manufacturers’ procedures. (section 2.4; lines 175-177..in revised version)

Determine the absorbance of extracted DNA at 260 nm using the Eppendorf BioSpectrometer® fluorescence with 1 μL of each sample, as described by the manufacturer (section 2.5; lines 179-180…in revised version).

Q6.    Figure 1 – needs to be moved to the Experimental design section with highlighted section that were optimized.

Answer: Done as suggested

Q7.    Results section: remove lines 164-167 (move into Introduction or Discussion)

Answer: We have moved the paragraph to the Introduction (lines 45-46, 60-62..in revised version).

Q8.  Table 1 – no significant difference in yield increase compared to QIAGEN method. 

Answer: Done as suggested. We have included statistical analysis in Table 1.

Q9.  Page 6, line 190 – I don’t see a symmetrical peak in Fig 2A for 'O_NP'  part of it. It is actually more asymmetrical compared to 'CTAB' or 'QIAGEN'

Answer: We apologize for this mistake and corrected the original sentence (line 201-203) as follows: (line 241-244 in revised version)

“The extracted gDNA obtained from the optimized protocol had an A260/A280 ratio of 1.87 and the absorbance showed a single peak at 260 nm (Table 1, Figure 2A), indicating that the gDNA was high molecular weight, pure and contamination-free.”

Q10. Explain 'After cleaning up using Trimmomatic'  - what did you clean and how?

Answer: We have corrected the original sentence (line 205-206) as follows: (line 258-260… in revised version) as follows:

“We remove low quality reads and trimming low quality bases using Trimmomatic v. 0.33 software [19] and assess the quality of our sequenced data.”

Q11.  Table 2 – need to add comparison with CTAB and QIAGEN methods.

Answer: Figure 2 showing the clear evidence that our proposed method produces a good quality gDNA compared to CTAB and QIAGEN methods, which is our main objective. In addition, the extraction protocol influenced the quality of gDNA and, in turn, the sequencing quality. NGS Library prepared using low quality gDNA (obtained using CTAB method) results inconsistent and unreliable SNP data (please see reference 3). We apologize that, owing to high cost, it is not possible for us to perform NGS using gDNA obtained via the CTAB and QIAGEN methods.

Q12.  Figure 3 - need to perform NGS using gDNA obtained via CTAB and QIAGEN methods and compare the sequence quality results for all 3 DNA extraction methods. Currently there is no comparison and no solid evidence that this optimized protocol is any better than QIAGEN’s or any other.

Answer: Figure 2 showing the clear evidence that our proposed method produces a good quality gDNA compared to CTAB and QIAGEN methods, which is our main objective. In addition, the extraction protocol influenced the quality of gDNA and, in turn, the sequencing quality. NGS Library prepared using low quality gDNA (obtained using CTAB method) results inconsistent and unreliable SNP data (please see reference 3). We apologize that, owing to high cost, it is not possible for us to perform NGS using gDNA obtained via the CTAB and QIAGEN methods.

.

Q13.  Page 8, line 238 – explain parameters: SLIDINGWINDOW, TRAILING, MINLEN.

Answer: Very few research articles explain detail of parameter meaning one by one, especially in case of very general tool like Trimmomatic [over 4000 citations since 2014]. We think readers easily access this parameter information.

Q16.  Show the evidence that this optimized protocol allows to save on cost of reagents and amounts of leaf tissue used (Lines 181-182).

Answer:  We did not propose our optimized nuclear plate method save reagents and leaf tissue (line 181-182). The amount of reagent we used here for 1 g leaf sample. Therefore, this protocol allows downscaling of regents for small quantity of leaf tissue (for example 100 mg). Therefore, we correct the paragraph as follows:

“Due to the high gDNA yield, this protocol allows a downscaling of the amounts of leaf tissue and reagents used” (lines 232-233---revised version).

Reviewer 4 Report

The authors optimized nuclear pellet (NP) method for extracting genomic DNA suitable for next generation sequencing resulting in good quality short reads. 

I have 3 major points regarding this paper:

- The obtained NP_DNA was not tested, by qPCR for example, for the nuclei purity level.

- The phenolics/polysaccharides contaminants of NP_DNA should also be monitored by the absorbance ratio at 260/230. This data should be included in Table 1.

- It would have been really interesting if NP_DNA was tested with third generation sequencing for long reads.

Other minor points are listed below:

L43: Delete "high accuracy" or re-phrase. NGS have high error frequency necessitating high coverage. 

L55: ...is expensive. I would add … is expensive and not suitable for all species.

L60: add polysaccharides to the list.

L77: … sodium hypochlorite, rinsed with water and incubated ...

L105: TE buffer contains 1 mM EDTA. Use TE-1 or other indication instead. 

L197: As mentioned above, add A260/A230 data in the table 1.

L206: Delete "high molecular weight"

L208: change … a clear, intact... to … a clear, high molecular weight band with little...   

L213: change "The DNAs" to The genomic DNA samples...

L214: change "the DNAs" to samples...

L222: change "in lots of pairs" to "in high number of pairs"

L223: change "high yields of reads" to "high read yields"

L225: change "sequencing and accurately" to sequencing assembly and accurate"

L226: change "regions where are" to "regions that are"

L230: change the whole sentence "These technologies is sensitive" to "These technologies require high quality intact DNA with high molecular weight."

L231:  change the whole sentence "....these sequences...." to "...these sequencing platforms better than conventional DNA extraction methods and will improve analytical genomics of any plant species.

L241: Add the name of the marker and company.

L244-L246: I have no idea what this text is related to. 

Author Response

Thank you for all of your detailed comments and suggestions. We found them useful as we approached our revision. You will find our responses to each of your points and suggestions. We are grateful for the time and energy you expended on our behalf.

Major comments:

Comment #1: The obtained NP_DNA was not tested, by qPCR for example, for the nuclei purity level.

Answer: Thank you for your comment. Following your suggestion, we have quantified nuclear, chloroplast and mitochondrial DNAs by qPCR and the data have included in the revised ms. (lines 186-196; 224-227, 270-296).

Comment #2: The phenolics/polysaccharides contaminants of NP_DNA should also be monitored by the absorbance ratio at 260/230. This data should be included in Table 1.

Answer: Thank you for your comment. We have monitored the absorbance ratio of the DNA extracted by each method at 260/230 and included the data in Table 1 (line 211-216).

Comment #3: It would have been really interesting if NP_DNA was tested with third generation sequencing for long reads.

Answer: Thank you for your comment.  Figure 2 showing the clear evidence that our proposed method produces a very good quality gDNA compared to CTAB, QIAGEN and NP methods, which is our main objective. Moreover, the DNAs obtained using the optimized NP method passed the stringent quality control requirements for the Illumina Hiseq X Ten platform and resulting good quality read yields. Based on the quality assessment data, we propose the DNAs obtained using our protocol will also be suitable for third generation sequencing for long reads.

Minor comments:

Comment #1: L43: Delete "high accuracy" or re-phrase. NGS have high error frequency necessitating high coverage. 

Answer: We consider this phrase "high accuracy" is correct for the following reasons: Current Illumina short read sequencers generate reads with highly accurate base calling. Our result (Figure 4) also shows the highly accurate base calling with high Phred-quality score (over 30, i.e. more than 99.9% reliability).  In the resequencing, highly accurate SNPs/INDELs can be detected with about 30x coverages depth (not so high). Indeed, by resequencing, many reference genomes have been corrected in the past decade.

Comment #2: L55: ...is expensive. I would add … is expensive and not suitable for all species.

Answer: Done as suggested. We replaced “is expensive to “is expensive and not suitable for all species” (line 57-58)

Comment #3: L60: add polysaccharides to the list.

Answer: Added as suggested (line 62)

Comment #4: L77: … sodium hypochlorite, rinsed with water and incubated ...

Answer: As your suggestion, we added “rinsed with water” in the revised ms. (line 80-81)

Comment #5: L105: TE buffer contains 1 mM EDTA. Use TE-1 or other indication instead. 

Answer: Thank you for your comment. We think reader can easily point out the TE buffer preparation through reading the guidelines presented in this manuscript. All authors decided to keep the original indication as ‘TE’.

Comment #6: L197: As mentioned above, add A260/A230 data in the table 1.

Answer: As per your suggestion, we included A260/A230 data in the Table 1. (line 211-216)

Comment #7: L206: Delete "high molecular weight"

Answer: We deleted the text "high molecular weight" from ms. (line 207-208)

Comment #8: L208: change … a clear, intact... to … a clear, high molecular weight band with little...   

Answer: Done as suggested. We changed “a clear, intact” to “a clear, high molecular weight band with little(line 218-219).

Comment #9: L213: change "The DNAs" to The genomic DNA samples...

Answer: Done as suggested. We changed “The DNAs” to “The genomic DNA samples” (line 299).

Comment #10: L214: change "the DNAs" to samples...

Answer: Done as suggested. We changed “The DNAs” to “samples” (line 300).

Comment #11: L222: change "in lots of pairs" to "in high number of pairs"

Answer: Done as suggested. We changed “in lots of pairs” to “in high number of pairs” (line 307-308).

Comment #12: L223: change "high yields of reads" to "high read yields"

Answer: Done as suggested. We changed “high yields of reads” to “high read yields” (line 309).

Comment #13: L225: change "sequencing and accurately" to sequencing assembly and accurate"

Answer: Done as suggested. We changed “sequencing and accurately” to “sequencing assembly and accurate(line 311).

Comment #14: L226: change "regions where are" to "regions that are"

Answer: Done as suggested. We changed “regions where are” to “regions that are” (line 312).

Comment #15: L230: change the whole sentence "These technologies is sensitive" to "These technologies require high quality intact DNA with high molecular weight."

Answer: Done as suggested. We replaced the whole sentence “These new sequencing technologies is sensitive to DNA quality, especially high-molecular weight DNA without smear is key to above applications to “These technologies require high quality intact DNA with high molecular weight.” (line 315-316).

Comment #16: L231:  change the whole sentence "....these sequences...." to "...these sequencing platforms better than conventional DNA extraction methods and will improve analytical genomics of any plant species.

Answer: Done as suggested. We replaced the whole sentence “these sequencers more closely than conventional DNA extraction methods, and it provides a powerful analytical opportunity for rice genomics to “these sequencing platforms better than conventional DNA extraction methods and will improve analytical genomics of any plant species(line 317-319).

Comment #17: L241: Add the name of the marker and company.

Answer: Added as suggested (line 252-253)

Comment #18: L244-L246: I have no idea what this text is related to. 

Answer: These texts correspond to the quality control results of genomic DNA samples performed by the Macrogen for NGS. We have deleted the following text “DNA quality control based on measurement of absorbance (A260/A280 ratio), quantification of DNA using Picogreen method using Victor X2® fluorometry and 1.0% agarose gel electrophoresis” to avoid any misunderstanding by readers.

Reviewer 5 Report

Authors propose one optimized method for extracting next generation sequencing (NGS) quality genomic DNA, from fresh leaf tissue of rice.

Comments to this study:

i)      The authors performed the extraction of genomic DNA with three methods, CTAB-based method, DNeasy plant Mini kit (Qiagen) method and nuclear pellet optimized method. In the experimentation we feel that authors should compare also the nuclear pellet optimized method with original nuclear pellet method. The differences between them are not clear and the authors should confirm the efficiency of the optimized method.

ii)     The authors performed the extraction of genomic DNA with three methods, however they only carry out downstream quality analyzes with the nuclear pellet optimized method. We think that they should compare all the tested methods in downstream quality analyzes as a way of demonstrating the efficiency of the proposed method (Table 2, page 9).

iii)    DNA quality was estimated by measuring the absorbance at 260 and 280 nm. However the A260/A230 absorbance ratio should also be considered. Readings at 230 nm wavelengths measure the concentration of salts, carbohydrates and other contaminants.

iv)   In figure 2B (page 8) it is not clear the DNA smearing of CTAB-based and Qiagen methods.

v)    Authors should standardize the way as they write volume units, sometimes they write ul (ex. page 3 line 87), ml (ex. page 4, line 126), and other times they write uL (ex. page 3, line 106; page 5, line 176).

vi)   Please avoid the use of abbreviations (PCI and CIA, page 1, line 24) in abstract. 

vii)  We suggest the change of the name of section 2. “Experimental design” to “materials and methods”. In our opinion, the section 2.3 “Optimization of nuclear pellet method” should be inside the section 2.2 “genomic DNA extraction”, since is one of the DNA extraction studied methods.

viii) Page 7, lines 190 – 193. The sentence are not results but rather information that must be contained in material and methods.

ix)   Page 7, line 203. The sentence are not results but rather information that must be contained in material and methods.

x)    Page 5, line 172. Space between words “method” and “using”.

xi)   Page 5, sections 2.4 and 2.5. Please review the text and correct verbal form.

Author Response

Thank you for all of your detailed comments and suggestions. We found them useful as we approached our revision. You will find our responses to each of your points and suggestions. We are grateful for the time and energy you expended on our behalf.

Comment #1: The authors performed the extraction of genomic DNA with three methods, CTAB-based method, DNeasy plant Mini kit (Qiagen) method and nuclear pellet optimized method. In the experimentation we feel that authors should compare also the nuclear pellet optimized method with original nuclear pellet method. The differences between them are not clear and the authors should confirm the efficiency of the optimized method.

Answer: Thank you for your suggestion. We have extracted the gDNA using original NP method and included data in revised ms. The Table 1, Figure 2 and Figure 3 now contain yield and quality comparison data of extracted DNA using the CTAB, QIAGEN, Original NP and optimized NP protocols (lines: 211-216; 230-252; 269-296).

Comment #2: The authors performed the extraction of genomic DNA with three methods, however they only carry out downstream quality analyzes with the nuclear pellet optimized method. We think that they should compare all the tested methods in downstream quality analyzes as a way of demonstrating the efficiency of the proposed method (Table 2, page 9).

Answer: Thank you for your comment. Figure 2B showing the clear evidence that our proposed method produces a good quality gDNA compared to the CTAB and QIAGEN and original NP methods, which is our main objective. In addition, the extraction protocol influenced the quality of gDNA and, in turn, the sequencing quality. NGS Library prepared using low quality gDNA (obtained using CTAB method) results inconsistent and unreliable SNP data (please see reference 3). We apologize that, owing to high cost, it is not possible for us to perform NGS using gDNA obtained via the CTAB and QIAGEN and original NP methods.

Comment #3: DNA quality was estimated by measuring the absorbance at 260 and 280 nm. However the A260/A230 absorbance ratio should also be considered. Readings at 230 nm wavelengths measure the concentration of salts, carbohydrates and other contaminants.

Answer: Thank you for your suggestion. We have monitored the absorbance ratio of the DNA extracted by each method at 260/230 and included the data in Table 1 (line: 211-216).

Comment #4: In figure 2B (page 8) it is not clear the DNA smearing of CTAB-based and Qiagen methods.

Answer: Thank you for your comment. We have revised the figure 2B in revised ms. which clearly shows the DNA smearing of CTAB-based, QIAGEN and original NP protocols (line: 238-246)

Comment #5: Authors should standardize the way as they write volume units, sometimes they write ul (ex. page 3 line 87), ml (ex. page 4, line 126), and other times they write uL (ex. page 3, line 106; page 5, line 176).

Answer: Thank you for your suggestion. We have checked all the volume units and revised the ms. whenever it corresponds (lines: 109-110; 180; 182). 

Comment #6: Please avoid the use of abbreviations (PCI and CIA, page 1, line 24) in abstract

Answer: Done as suggested. We changed the texts; PCI to phenol/chloroform/isoamyl alcohol (25:24:1) and CIA to chloroform/isoamyl alcohol (24:1) in abstract (line: 25). 

Comment #7: We suggest the change of the name of section 2. “Experimental design” to “materials and methods”. In our opinion, the section 2.3 “Optimization of nuclear pellet method” should be inside the section 2.2 “genomic DNA extraction”, since is one of the DNA extraction studied methods.

Answer: Done as suggested (line: 78)

Comment #8: Page 7, lines 190 – 193. The sentence are not results but rather information that must be contained in material and methods.

Answer: Done as suggested. The sentences “We extracted total gDNA from 20-day-old leaf tissues of the rice variety, Yukinko-mai, using the traditional CTAB-based method, the QIAGEN DNeasy Plant Mini Kit and the optimized NP method. The quantity of the genomic DNA was tested using the Qubit dsDNA HS Assay Kit (Thermofisher Scientific), which is very reliable [15]” has been deleted from result section and technically explained in M&M section.

Comment #9: Page 7, line 203. The sentence are not results but rather information that must be contained in material and methods.

 Answer: Done as suggested. The sentence “We determined DNA absorbance at different wavelengths using Eppendorf BioSpectrometer® fluorescence.” has been deleted from result section and technically explained in M&M section.

Comment #10: Page 5, line 172. Space between words “method” and “using”.

 Answer: Done as suggested. We have revised the entire sentence according to your comment#11

Comment #11: Page 5, sections 2.4 and 2.5. Please review the text and correct verbal form.

 Answer: Done as suggested. We have revised the section 2.4 “Quantify the DNA isolated by each methodusing QubitTM dsDNA HS Assay Kit, following the manufacturers’ procedures” to

“The quantity of genomic DNA isolated by each method was tested with a QubitTM dsDNA HS Assay Kit, following the manufacturers’ procedures.” (line 176-177)

and section 2.5 “Determine the quality of extracted DNA at 260 nm using the Eppendorf BioSpectrometer® fluorescence with 1 μL of each sample, as described by the manufacturer. Use agarose gel electrophoresis to further assess the gDNA quality, in a 0.8 % agarose gel visualizing the DNA using 177 0.5 μg/mL ethidium bromide in TBE” to

“The quality of extracted DNA was determined at 260 nm using the Eppendorf BioSpectrometer® fluorescence with 1 µl of each sample, as described by the manufacturer. The agarose gel electrophoresis was used to further assess the gDNA quality, in a 0.8 % agarose gel visualizing the DNA using 0.5 µg/ml ethidium bromide in TBE” (line 179-182).

Round 2

Reviewer 2 Report

The authors have addressed all the comments

Author Response

 Thanks for your comments.

Reviewer 3 Report

Considering the the revised manuscript, I have not seen any significant improvements of the originally submitted paper.  There is no evidence of significant optimization of the existing gDNA extraction protocols, that would lead to significantly reduced time for DNA extraction, much higher yields or much lower costs. Incorporated optimization steps are minimal, such as use of Proteinase K, RNAse A, PCI and CIA, and have been known and used in plant DNA extractions for years already.  Again, the data provided is Figure 3 is not compared to the same NGS results for rice obtained while using established gDNA extraction methods. There is no strong evidence that this protocol is any different/better compared to the ones that are already being successfully used.

Author Response

 Thanks for your comments.

Reviewer 5 Report

Dear authors

We suggest the following minor alterations:

i) Abstract: include authority in scientific names

ii) lines 68 and 71: include scientific names and authority. 

iii) Table 1: include rice scientific name.

iv) Figure 2, 3 and 4, and table 2: indicate that is DNA from rice (Oryza sativa L.) plant. 

v) In Figure 2, use small not capital letter for "kb" DNA ladder.

Author Response

Thank you for all of your detailed comments and suggestions. We found them useful as we approached our revision. You will find our responses to each of your points and suggestions. We are grateful for the time and energy you expended on our behalf.

Comment #1: Abstract: include authority in scientific names

Answer: Thank you for your suggestion. We have included the authority in scientific names (lines: 21-22).

Comment #2: lines 68 and 71: include scientific names and authority. 

Answer: Done as suggested. We included scientific names and (lines: 68; 71).

Comment #3: Table 1: include rice scientific name.

Answer: Done as suggested. We included scientific name and authority (line: 211).

Comment #4: Figure 2, 3 and 4, and table 2: indicate that is DNA from rice (Oryza sativa L.) plant.

Answer: Done as suggested. We included the appropriate sentence along with the indication “DNA from rice (Oryza sativa L.) plant” in Figure 2, 3 and 4 and Table 2. (Lines: 248; 293-294; 324; 330-331).

Comment #5: In Figure 2, use small not capital letter for "kb" DNA ladder.

Answer: Done as suggested. We changed the text; ‘Kb’ to ‘kb’ (line: 253).